# PERSONAEVAL: BENCHMARKING LLMs ON ROLE-PLAYING EVALUATION TASKS

## ABSTRACT

Role-playing in large language models (LLMs) has become a crucial area of research, enabling models to simulate diverse personas and tailor responses, significantly impacting natural language understanding and human-computer interaction. However, while advanced LLMs like GPT-4 are used to evaluate role-playing methods, their reliability in providing accurate assessments remains uncertain, especially in distinguishing nuanced role-playing characteristics. In this paper, we introduce PersonaEval, a benchmark designed to assess the effectiveness of LLMs in role-playing evaluation tasks. We frame the problem as a classification task to determine whether an LLM evaluator can distinguish between sentences from different levels of expertise based solely on linguistic cues. Using real-world data from the Wired 5 Levels video series—where experts explain concepts to five distinct audiences: a child, a teenager, a college student, a graduate student, and another expert—we design three evaluation settings that correspond to commonly used LLM evaluation approaches: single answer role grading, pairwise role comparison, and reference-guided role grading. These settings aim to capture various aspects of how effectively LLMs evaluate role-playing performance. Our study highlights the limitations of current LLMs in persona evaluation tasks and underscores the need for further research to enhance their evaluation capabilities. We provide a foundation for future work aimed at improving the accuracy and professionalism of LLM evaluators in role-playing contexts.

## 1 INTRODCUTION

Role-playing has rapidly developed into a crucial area in the research and application of large language models (LLMs) (Shao et al., 2023; Tao et al., 2023; Lu et al., 2024; Xu et al., 2024). The ability of LLMs to simulate diverse personas and adapt their responses accordingly holds significant implications for natural language understanding, human-computer interaction, and various downstream tasks (Tseng et al., 2024; Chen et al., 2024a;b). Most current research on role-playing employs state-of-the-art LLMs, such as GPT-4 (OpenAI, 2023), to evaluate different role-playing methods and models (Tang et al., 2024; Wang et al., 2024a). These advanced LLMs are often presumed capable of accurately assessing whether a role player's output aligns with the target role.

However, the reliability of LLM evaluators in providing correct assessments remains uncertain. Some studies have begun to investigate this issue, such as testing the evaluator's instruction-following abilities (Murugadoss et al., 2024; Wei et al., 2024) or task feasibility (Zhang et al., 2024). Yet, research specifically focused on evaluating LLMs' effectiveness in role-playing evaluation is still limited. Can LLMs accurately distinguish user levels or make nuanced judgments between different roles? Moreover, can their evaluations, even with reference guidance, be trusted to reflect true role consistency and authenticity? Despite their widespread use for such tasks, more research is needed to explore their accuracy and professionalism as role-playing evaluators.

To address this gap, we propose a benchmark called PersonaEval to evaluate LLMs effectiveness in role-playing evaluation tasks, as shown in Figure 1. Framed as a classification problem, our benchmark assesses whether an LLM evaluator can reliably distinguish between sentences originating from different levels of expertise. The goal is to determine if the LLM can accurately classify these sentences based solely on linguistic cues, providing a deterministic approach to evaluate its capability in identifying nuanced role-playing characteristics.

Figure 1: **Our proposed benchmark PersonaEval, constructed on real-world data, is to evaluate LLMs effectiveness in role-playing evaluation tasks.** The content of the video series is that experts explain specific concepts to five distinct audiences, including child, teenager, college student, graduate student, and another fellow expert. LLMs are required to capture the linguistic nuances at each level to recognize and evaluate diverse communication styles and complexities.

To construct our benchmark, we leveraged real-world data from the popular Wired 5 Levels video series[1]. In this series, experts explain complex concepts to five distinct audiences: a child, a teenager, a college student, a graduate student, and another fellow expert. The series is renowned for its ability to adapt explanations to varying levels of prior knowledge, making it an ideal source for evaluating the capability to adjust language complexity.

Our PersonaEval encompasses 26 diverse topics and includes 130 unique speakers, resulting in dialogues that average over 17 turns of interaction. By focusing on these specific roles, we aim to assess the nuanced understanding and discriminatory abilities of LLMs in tailoring language use across different age groups and educational backgrounds. Furthermore, the dialogues in our benchmark are structured and goal-oriented rather than random exchanges. Each conversation is centered on explaining a concept at varying levels of complexity, meticulously tailored to match the audience's understanding. This setup presents a meaningful and structured challenge for LLMs, testing their ability to modulate explanations appropriately. Since the roles in the "5 Levels" series represent common personas, the LLMs evaluate familiar character archetypes. This familiarity reduces the risk of compromised performance due to unfamiliarity with the roles, allowing for a more accurate assessment of the models' capabilities in language adaptation and audience-specific communication.

To comprehensively evaluate the performance of LLMs in role-playing tasks using our PersonaEval benchmark, we designed three distinct evaluation settings: single answer role grading, pairwise role comparison, and reference-guided role grading. These settings correspond to established evaluation methodologies for LLMs (Zheng et al., 2023): single-answer grading, pairwise comparison, and reference-guided evaluation, respectively. In the single answer role grading setting, we assess the model's ability to recognize and categorize user inputs across different proficiency levels, testing its nuanced understanding of varied user interactions. The pairwise role comparison setting examines how effectively the model can distinguish between different roles, highlighting its adaptability to diverse contextual personas. In the reference-guided evaluation, we introduce reference examples to guide the LLM's judgments, evaluating its capacity to leverage prior examples to enhance evaluation accuracy. By employing these varied evaluation settings, we aim to capture multiple facets of how LLMs can effectively assess role-playing performance.

Our study reveals significant limitations in current LLMs when tasked with evaluating role-playing performances, underscoring the need for continued research to enhance their evaluative capabilities. The proposed PersonaEval benchmark serves as a foundational tool for future efforts aimed at improving the professionalism and precision of LLM evaluators in role-playing contexts. By addressing these limitations, we contribute to the advancement of models that are proficient not only in generating role-specific content but also in evaluating such content with a high degree of accuracy and reliability.

---

[1]https://www.wired.com/video/series/5-levels

## 2 RELATED WORK

**Role-playing Evaluation**   The use of LLMs as evaluators in role-playing tasks has gained traction due to their efficient and scalable capabilities in assessing performance, providing feedback, and aiding in decision-making processes (Gusev, 2024; Wang et al., 2024b). Three primary evaluation approaches have emerged in this context: single-answer grading, pairwise comparison, and reference-guided grading (Zheng et al., 2023).

For the single answer grading, an LLM evaluates a single response and directly assigns a score based on predefined metrics such as coherence, role alignment, or content accuracy. MMRole (Dai et al., 2024) introduces a reward model to better align LLM-generated evaluations with those of human judges. Pairwise comparison is commonly used to judge subtle differences between responses from two baseline models and identify which one better meets the intended criteria. RoleLLM (Wang et al., 2024a) uses GPT-4 to compare and rank generated samples, determining metrics such as win rates and average rankings for baseline models to evaluate instruction following and role generalization. Reference-guided grading provides a reference solution to the LLM evaluator. Neeko (Yu et al., 2024) leverages GPT-3.5 as a judge, prompting step-by-step scoring of dialogue performance based on predefined metrics.

These approaches aim to assess how effectively LLMs play specific roles, contributing to the overall understanding of their performance in role-playing scenarios. However, is the evaluating results reliable and stable, challenges remain in ensuring that the evaluations are reliable and accurately reflect the nuances of role adherence. Current approaches to using LLMs in role-playing evaluations are not without flaws.

**Evaluating LLM Evaluator**   The reliability of LLMs as evaluators is gaining increasing attention in the community (Son et al., 2024; Wei et al., 2024; Zhang et al., 2024). Several studies test the instruction-following abilities of LLMs in their role as evaluators. Judging LLM-as-a-Judge (Zheng et al., 2023) addresses this issue by using advanced LLMs to assess performance on open-ended questions. To verify alignment between LLM judgments and human preferences, they introduce two benchmarks: MT-Bench, a multi-turn question set, and Chatbot Arena, a crowdsourced battle platform. Wang et al. (2023) approach ChatGPT as a human-like evaluator by providing task-specific (e.g., summarization) and aspect-specific (e.g., relevance) instructions to prompt ChatGPT for evaluating the outputs of various NLG (Natural Language Generation) models. They conduct experiments on five NLG meta-evaluation datasets, covering tasks like summarization, story generation, and data-to-text conversion. Evaluating the Evaluator (Murugadoss et al., 2024) explores whether LLM assessments are based purely on prompt instructions or also reflect inherent preferences for high-quality data similar to their fine-tuning data. Their dataset spans tasks such as text summarization, conversation quality, task solution quality, and generated story quality.

However, despite these explorations, the evaluation of LLMs specifically as role-playing task evaluators remains underexplored. Current work often lacks the focus on assessing the nuanced abilities of LLMs to differentiate role levels and align with role-specific contexts, highlighting a gap in understanding LLMs' capacity for accurate role-play assessment.

## 3 PERSONAEVAL

In this paper, we introduce PersonaEval, a novel benchmark for evaluating the capacity of LLMs to act as role-playing evaluators. Our objective is to assess how effectively LLMs can classify and distinguish specific roles based on language usage associated with different roles. This classification capability is crucial for determining whether role-playing evaluators can discern contextual differences in interactions across a diverse range of audiences.

### 3.1 BENCHMARK CONSTRCTUION

Our PersonaEval is derived from the Wired "5 Levels" video series, where experts explain complex concepts—such as quantum computing, blockchain, and artificial intelligence—to five distinct audiences: a child, a teenager, a college student, a graduate student, and another fellow expert. This pedagogical approach creates a rich structure that spans a wide range of comprehension levels, vo-

Table 1: **Summary of dialogue characteristics across varying audience expertise levels in the proposed PersonaEval.** The benchmark consists of 130 dialogues, categorized into five distinct roles, with each role contributing 26 dialogues. These dialogues encompass a range of audience expertise levels, offering a robust resource for analysis and evaluation.

|  | Child | Teen | College Student | Grad Student | Expert | Overall |
|---|---|---|---|---|---|---|
| Avg Turns[†] | 22.54 | 16.96 | 16.46 | 16.27 | 17.08 | 17.86 |
| Min/Max Turns | 7/51 | 5/30 | 5/31 | 6/29 | 8/38 | 5/51 |
| Avg Tokens | 585.23 | 786.46 | 929.58 | 899.65 | 1239.88 | 888.16 |
| Min/Max Tokens | 152/989 | 191/1624 | 324/1517 | 258/1476 | 455/2786 | 152/2786 |

† Two turns constitute a complete back-and-forth conversation.

cabulary, and cognitive abilities. Such diversity provides a valuable foundation for evaluating the capability of LLMs to adapt their role-specific behaviors when interacting with various conversational partners. The primary focus is not only on the LLM's ability to embody a designated role but also on its proficiency in personalizing responses by adjusting complexity, tone, and content to match the listener's level of understanding.

Our PersonaEval encompasses 130 unique characters, each engaging in dialogues that average over 17 rounds of interaction per conversation, as detailed in Table 1. This extensive compilation facilitates a comprehensive exploration of both static role adherence—how consistently an LLM maintains its character—and dynamic personalization—how the LLM tailors its responses based on the listener's profile. The five distinct audience levels represent varying degrees of conversational complexity, allowing us to assess the LLM's ability to simplify explanations for a child or delve into highly technical discussions with an expert peer. This range provides a robust platform for evaluating the model's adaptability and responsiveness across diverse conversational contexts.

## 3.2 EVALUATION SETTING

To systematically assess the effectiveness of LLM evaluators in role-playing scenarios, we have designed three benchmark settings, as shown in Figure 2. Each setting corresponds to a common use case of LLM evaluators, as identified by Zheng et al. (2023): (1) Single Answer Grading: LLM judger assigns a score to a single response; (2) Pairwise Comparison: LLM judger is presented with a question and two possible answers, determining which is better or if they are equally suitable; (3) Reference-Guided Grading: LLM evaluates a response against a provided reference solution when applicable.

### 3.2.1 SINGLE ANSWER ROLE GRADING

The first evaluation setting involves a five-level classification task. The LLM is presented with several turns of dialogue and must determine to which level the responses correspond: child, teenager, college student, graduate student, or expert. This task aligns with the single-answer grading approach, requiring the LLM to assign appropriate classifications based on the input, akin to accurately scoring a response.

We employ the PersonaEval benchmark, a dataset specifically designed to test the LLM's ability to discern varying levels of expertise in user input. By evaluating whether LLM evaluators can accurately identify the user's knowledge level from the context, we can assess the model's understanding of nuanced differences in language and content. The goal is to ensure that the LLM can tailor explanations appropriately for different audiences, enhancing the effectiveness of personalized communication.

### 3.2.2 PAIRWISE ROLE COMPARISON

The second setting focuses on pairwise role comparison, corresponding to the binary choice evaluation approach. In this scenario, the LLM is given two responses generated from different roles and

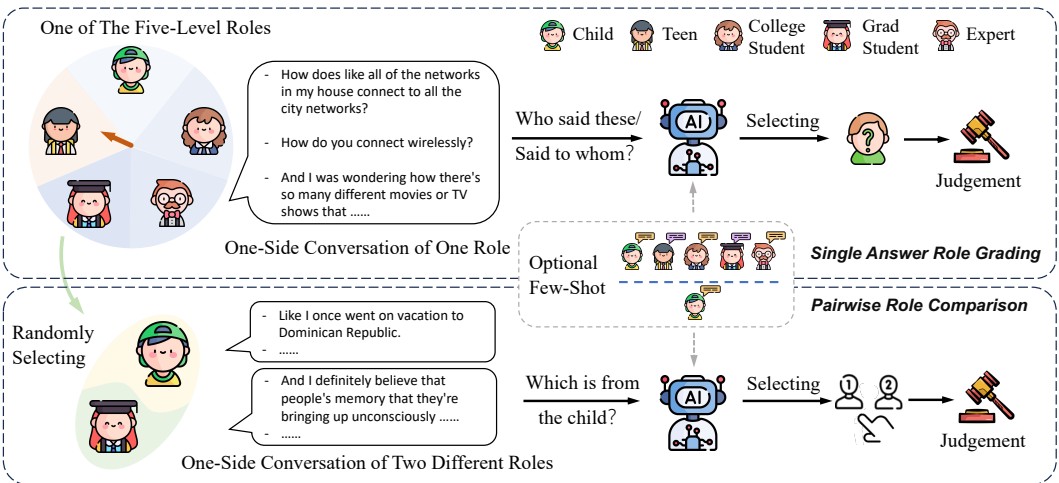

Figure 2: **Evaluation settings for assessing role-playing evaluators.** The single answer role grading task analyzes LLMs' ability to categorize one-sided conversations into five distinct levels. The pairwise role comparison task assesses the model's accuracy in determining which of two role-based responses aligns with a specified target role. Few-shot learning offers reference examples to improve performance in both grading and comparison tasks.

must decide which response aligns more closely with a specified target role. For example, the model might compare an answer from a college student with one from an expert to determine which better fits the role of a college student.

This setup mirrors the pairwise comparisons commonly used to evaluate role-playing performance, requiring the LLM to discern subtle differences between responses. By making accurate judgments about the appropriateness of each response relative to the target role, we can evaluate the model's ability to maintain role consistency and provide contextually appropriate information.

### 3.2.3 REFERENCE-GUIDED ROLE GRADING

The third setting extends both the single answer role grading and pairwise role comparison tasks by incorporating a few-shot learning approach. Here, we provide reference examples to the LLM to aid in evaluation. By including examples that demonstrate correct classifications or comparisons, the LLM gains additional context to refine its judgments.

This reference-guided setting leverages in-context learning to improve performance. By exposing the model to exemplars, we aim to enhance its understanding of the nuances associated with each role. Consequently, the LLM becomes better equipped to accurately classify levels or select the appropriate response in pairwise comparisons, leading to more reliable and consistent evaluations.

## 4 EXPERIMENTS

### 4.1 SETTINGS

We select a diverse set of state-of-the-art LLMs for evaluation, including GPT-4o OpenAI (2023) and GPT-3.5-turbo OpenAI (2023) from OpenAI, Qwen-Max, Qwen-Plus, and Qwen-Turbo from Qwen Family Bai et al. (2023), as well as the open-source model DeepSeek-v2.5 Bi et al. (2024). To assess the ability of these models to identify different roles, we conduct experiments on single answer role grading of five-level classification, which involves all mentioned models. For the pairwise role comparison and reference-guided role grading experiments, we focus on three representative models: GPT-4o, GPT-3.5-turbo, and Qwen-Max. The selection includes a top-performing model from both the GPT and Qwen families, with GPT-3.5-turbo included for vertical comparison. All the prompts used in the experiments are provided in the appendix.

## 4.2 SINGLE ANSWER ROLE GRADING OF FIVE-LEVEL CLASSIFICATION

Table 2: Classification Performance on Explainer Side of PersonaEval Benchmark

| model | Metric | 5 turns | | | | | | all turns | | | | | |
|---|---|---|---|---|---|---|---|---|---|---|---|---|---|
| | | Child | Teen | Undergrad | Grad | Expert | Average | Child | Teen | Undergrad | Grad | Expert | Average |
| GPT-3.5-Turbo | Precision | 97.1 | 21.8 | 19.2 | 26.5 | 35.4 | 40.0 | 100.0 | 18.8 | 25.3 | 27.6 | 30.2 | 40.4 |
| | Recall | 22.2 | 24.5 | 22.1 | 42.8 | 49.7 | 32.3 | 21.8 | 12.8 | 26.9 | 39.7 | 47.4 | 29.7 |
| GPT-4o | Precision | 91.0 | 45.4 | 32.0 | 30.7 | 45.2 | 48.9 | 98.6 | 71.4 | 41.5 | 32.2 | 47.3 | 58.2 |
| | Recall | 71.9 | 34.1 | 33.9 | 43.4 | 52.2 | 47.1 | 80.8 | 53.8 | 33.3 | 41.0 | 64.1 | 54.6 |
| Qwen-max | Precision | 96.3 | 25.6 | 27.4 | 36.9 | 56.9 | 48.6 | 100.0 | 25.7 | 30.8 | 35.5 | 40.0 | 46.4 |
| | Recall | 31.1 | 33.3 | 63.0 | 54.6 | 3.8 | 37.2 | 24.4 | 25.6 | 71.8 | 46.2 | 5.1 | 34.6 |
| Qwen-plus | Precision | 91.9 | 30.9 | 27.2 | 32.4 | 48.7 | 46.2 | 91.5 | 29.9 | 25.3 | 33.1 | 42.9 | 44.5 |
| | Recall | 57.4 | 27.7 | 52.4 | 45.7 | 21.0 | 40.8 | 43.6 | 19.2 | 43.6 | 51.3 | 25.6 | 36.7 |
| Qwen-turbo | Precision | 33.3 | 6.5 | 9.6 | 25.7 | 25.6 | 20.2 | 0.0 | 3.7 | 12.8 | 25.6 | 26.5 | 13.7 |
| | Recall | 0.2 | 2.4 | 16.5 | 61.1 | 32.8 | 22.6 | 0.0 | 1.3 | 14.1 | 66.7 | 28.2 | 22.1 |
| Deepseek-chat | Precision | 98.5 | 22.4 | 15.3 | 30.0 | 39.8 | 41.2 | 96.3 | 22.8 | 16.6 | 28.1 | 46.0 | 42.0 |
| | Recall | 35.0 | 20.0 | 20.7 | 65.2 | 28.4 | 33.9 | 34.6 | 12.8 | 15.4 | 61.5 | 44.9 | 33.8 |

Table 3: Classification Performance on Audience Side of PersonaEval Benchmark

| model | Metric | 5 turns | | | | | | all turns | | | | | |
|---|---|---|---|---|---|---|---|---|---|---|---|---|---|
| | | Child | Teen | Undergrad | Grad | Expert | Average | Child | Teen | Undergrad | Grad | Expert | Average |
| GPT-3.5-Turbo | Precision | 78.4 | 41.7 | 40.5 | 35.9 | 57.6 | 50.8 | 78.5 | 46.9 | 46.4 | 36.9 | 55.4 | 52.8 |
| | Recall | 52.0 | 50.8 | 39.7 | 26.3 | 91.5 | 52.1 | 37.2 | 47.4 | 42.3 | 32.1 | 96.2 | 51.0 |
| GPT-4o | Precision | 68.4 | 48.9 | 59.3 | 64.8 | 84.0 | 65.1 | 66.1 | 61.0 | 69.4 | 69.4 | 80.3 | 69.2 |
| | Recall | 93.6 | 36.0 | 53.0 | 54.0 | 76.9 | 62.7 | 92.3 | 44.9 | 62.8 | 64.1 | 83.3 | 69.5 |
| Qwen-max | Precision | 77.5 | 43.9 | 43.5 | 38.8 | 88.5 | 58.4 | 67.9 | 45.8 | 51.5 | 52.4 | 91.7 | 61.9 |
| | Recall | 63.6 | 49.2 | 72.7 | 49.8 | 25.1 | 52.1 | 35.9 | 47.4 | 82.1 | 70.5 | 44.9 | 56.2 |
| Qwen-plus | Precision | 74.7 | 47.7 | 48.4 | 64.5 | 82.8 | 63.6 | 69.3 | 53.2 | 66.0 | 75.3 | 76.9 | 68.1 |
| | Recall | 85.9 | 58.6 | 47.6 | 37.8 | 77.2 | 61.4 | 66.7 | 60.3 | 67.9 | 51.3 | 91.0 | 67.4 |
| Qwen-turbo | Precision | 79.3 | 30.8 | 34.7 | 29.1 | 40.0 | 42.8 | 51.1 | 39.5 | 37.6 | 37.3 | 34.2 | 40.0 |
| | Recall | 10.1 | 59.2 | 39.4 | 20.3 | 66.1 | 39.0 | 9.0 | 51.3 | 32.1 | 38.5 | 56.4 | 37.4 |
| Deepseek-chat | Precision | 83.1 | 41.3 | 52.7 | 46.9 | 82.7 | 61.3 | 83.0 | 50.5 | 61.8 | 56.0 | 70.3 | 64.3 |
| | Recall | 52.4 | 70.9 | 54.6 | 58.7 | 51.5 | 57.6 | 38.5 | 74.4 | 57.7 | 62.8 | 70.5 | 60.8 |

The five-level classification experiments are conducted from two perspectives: the explainer side and the audience side. For each perspective, we use two input settings: five turns of dialogue and all available dialogue turns from that side. This approach enables us to evaluate how well LLMs, acting as evaluators, can classify role levels based on linguistic cues from either the explainer's language or the language tailored to different audiences. By comparing performance across these settings, we aim to understand how the amount of dialogue context and the perspective taken affect the model's ability to accurately identify the correct role level in this multi-class classification task, shedding light on the LLMs' role-playing evaluation capabilities.

Tables 2 and 3 compare classification performance based on explainer-side and audience-side dialogues. For explainer-side dialogues (Table 2), models like GPT-4o show improvement when given full context, with macro-averaged precision rising from 48.9% to 58.2% and recall from 47.1% to 54.6%. However, while gpt-3.5-turbo excels in precision for "Child", its recall remains low across levels, and the Qwen models maintain moderate precision but struggle with recall, especially for the "Expert" role. DeepSeek-v2.5 performs well for "Child" but similarly falls short on recall for nuanced roles, indicating challenges in distinguishing subtleties using explanation-focused cues.

In contrast, audience-side language (Table 3) provides clearer distinctions, leading to better model performance overall. Full context enhances the precision and recall for all models, with GPT-4o reaching a macro average of 69% in both metrics, demonstrating adaptability to role-specific audience cues. Compared to explanation-side language, audience-side prompts yield higher precision and recall across models, as seen in the moderate but consistent gains for gpt-3.5-turbo and the Qwen models. DeepSeek also benefits from audience context, though its recall varies across roles. These results indicate that the language style directed at different audience levels is more informative for role classification, suggesting the value of considering the listener's perspective to improve LLM role-playing evaluations.

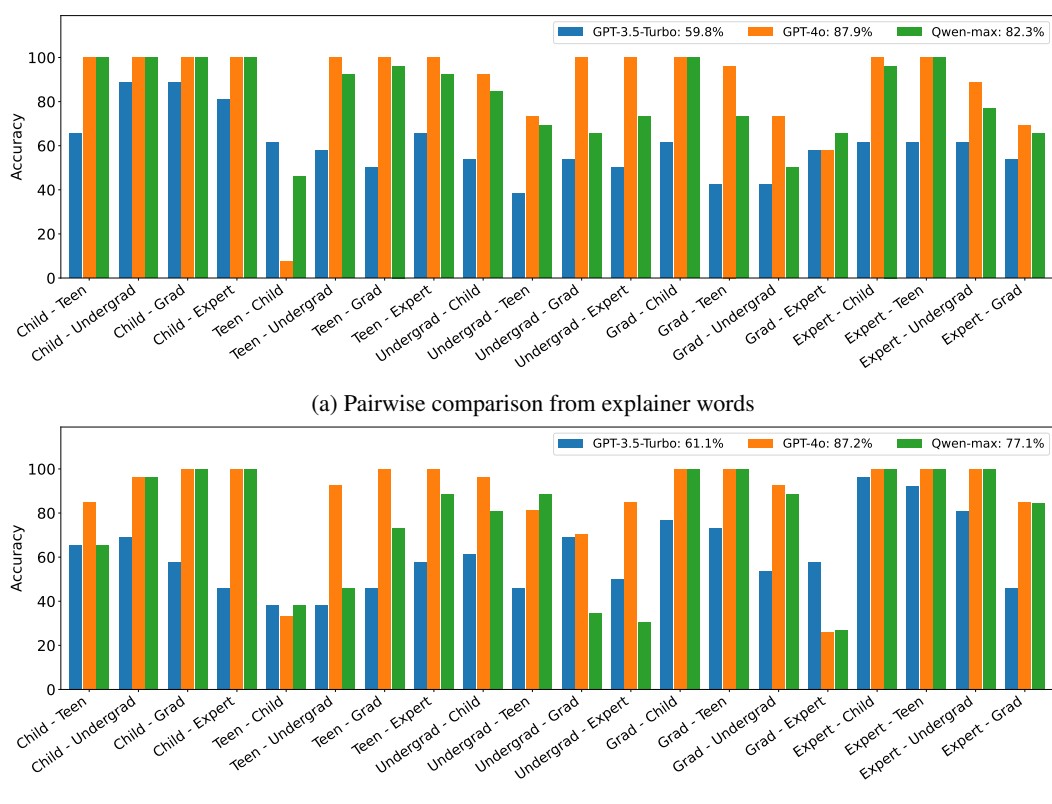

(a) Pairwise comparison from explainer words

(b) Pairwise comparison from audience words

Figure 3: **Pairwise role comparison performance of LLMs from explainer and audience perspectives.** The figure shows that GPT-4o outperforms other models in pairwise role comparisons, followed by Qwen-Max and GPT-3.5-turbo. All models excel with roles that have significant language differences (e.g., "Child - Expert") but struggle with similar levels (e.g., "Grad - Undergrad"), indicating challenges in evaluating comparable role-playing abilities.

These findings imply that the ability of LLMs to classify role levels effectively depends on the context available, with richer context leading to better performance. Audience-based classifications provide more accurate distinctions between role levels than explainer-based classifications, likely due to the clearer linguistic differences when addressing specific audiences. The results underscore the need for models to be trained to better understand audience-based cues to enhance their role-playing evaluation capabilities and improve their overall accuracy in role-specific language tasks.

We have also done the stability experiments, repeating each experiment for three times, and calculated the corresponding average and variance in Appendix A.

## 4.3 PAIRWISE ROLE COMPARISON IN 20 PERMUTATIONS

The pairwise role comparison experiments are designed to evaluate how effectively LLMs can identify dialogues matching a target level, similar to how an LLM evaluator compares responses from two models. In this setup, the LLM is provided with two sets of dialogue excerpts—each spoken by an explainer addressing audiences at different levels or by audiences responding to explanations tailored for different levels. The LLM then selects which excerpt best aligns with the given target level. Unlike the five-level classification that considers both five-turn segments and all dialogue turns, pairwise role comparison uses the full dialogue context from either the explainer or audience side, as the impact of context length has already been explored in the classification experiments.

Given the five possible levels, there are 20 permutations of role pairs, and we report both the individual results for each type and the overall average performance in Figure 3. The figure shows that GPT-4o consistently performs best in pairwise role comparison tasks, followed by Qwen-Max, and

Table 4: **Contradiction rates in pairwise role comparison across explainer and audience perspectives.** The use of permutations for comparisons leads to inconsistent results between opposite permutations of the same role pair, highlighting instability in LLM evaluations, which indicates limitations in the reliability of LLMs as evaluators of nuanced role distinctions.

| | Explainer Side | | | Audience Side | | |
|---|---|---|---|---|---|---|
| | GPT-3.5-Turbo | GPT-4o | Qwen-max | GPT-3.5-Turbo | GPT-4o | Qwen-max |
| Child - Teen | 34.6 | 92.3 | 53.8 | 73.1 | 73.1 | 65.4 |
| Child - Undergrad | 50.0 | 7.7 | 15.4 | 61.5 | 0.0 | 23.1 |
| Child - Grad Student | 42.3 | 0.0 | 0.0 | 50.0 | 0.0 | 0.0 |
| Child - Expert | 34.6 | 0.0 | 3.8 | 50.0 | 0.0 | 0.0 |
| Teen - Undergrad | 50.0 | 26.9 | 30.8 | 69.2 | 23.1 | 65.4 |
| Teen - Grad Student | 38.5 | 3.8 | 30.8 | 42.3 | 0.0 | 26.9 |
| Teen - Expert | 42.3 | 0.0 | 7.7 | 42.3 | 0.0 | 11.5 |
| Undergrad - Grad Student | 57.7 | 26.9 | 61.5 | 38.5 | 38.5 | 61.5 |
| Undergrad - Expert | 42.3 | 11.5 | 26.9 | 53.8 | 15.4 | 69.2 |
| Grad Student - Expert | 73.1 | 65.4 | 61.5 | 57.7 | 80.8 | 73.1 |
| Average | 46.5 | 23.5 | 29.2 | 53.8 | 23.1 | 39.6 |

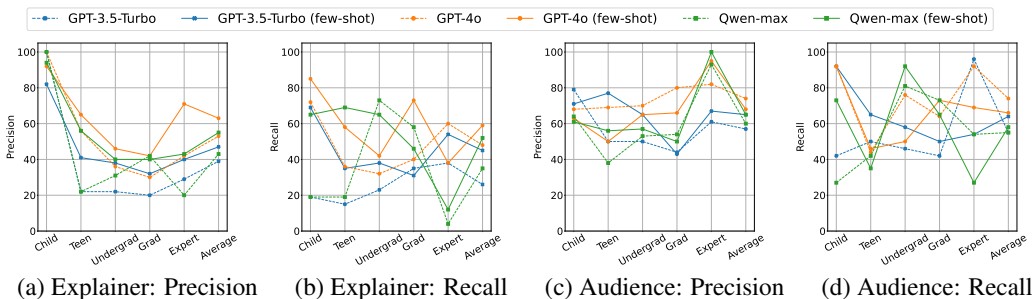

| (a) Explainer: Precision | (b) Explainer: Recall | (c) Audience: Precision | (d) Audience: Recall |

Figure 4: **Impact of few-shot learning on precision and recall across explainer and audience perspectives in five-level classification.** Few-shot examples notably boost precision and recall for GPT-4o, especially in distinct roles like "Child" and "Expert". In contrast, their impact on GPT-3.5-turbo and Qwen-Max is inconsistent, with some roles improving while others decline. This suggests that while few-shot learning enhances advanced models, its effects are less predictable for other LLMs, particularly in nuanced roles.

then GPT-3.5-turbo. A key observation is that all models achieve higher accuracy when comparing role pairs with significant language differences (e.g., "Child - Expert") but struggle with pairs that have closer role levels (e.g., "Grad - Undergrad"). This indicates that LLMs face challenges when evaluating models with similar role-playing abilities. Additionally, since the comparisons are based on permutations, opposite permutations of the same role pair do not always produce consistent results, revealing instability in LLM evaluations. Table 4 further explores these contradictions, highlighting inconsistencies in model judgments that suggest limitations in the LLMs' reliability as evaluators in nuanced role distinctions.

## 4.4 REFERENCE-GUIDED ROLE GRADING WITH FEW-SHOT LEARNING

To explore the influence of few-shot learning on the LLM evaluator task, we incorporate reference examples into the task prompts. We randomly select a set of paragraphs to serve as examples for the LLMs. In the case of five-level classification, we include five examples—one for each level—in the prompt. For the pairwise role comparison, we provide examples demonstrating the specific target level. This approach aims to assess how exposure to reference examples affects the model's performance in identifying and comparing role levels.

Figures 4 and 5 illustrate the impact of few-shot learning on LLM performance in five-level classification and pairwise role comparison tasks, respectively, across both explainer and audience per-

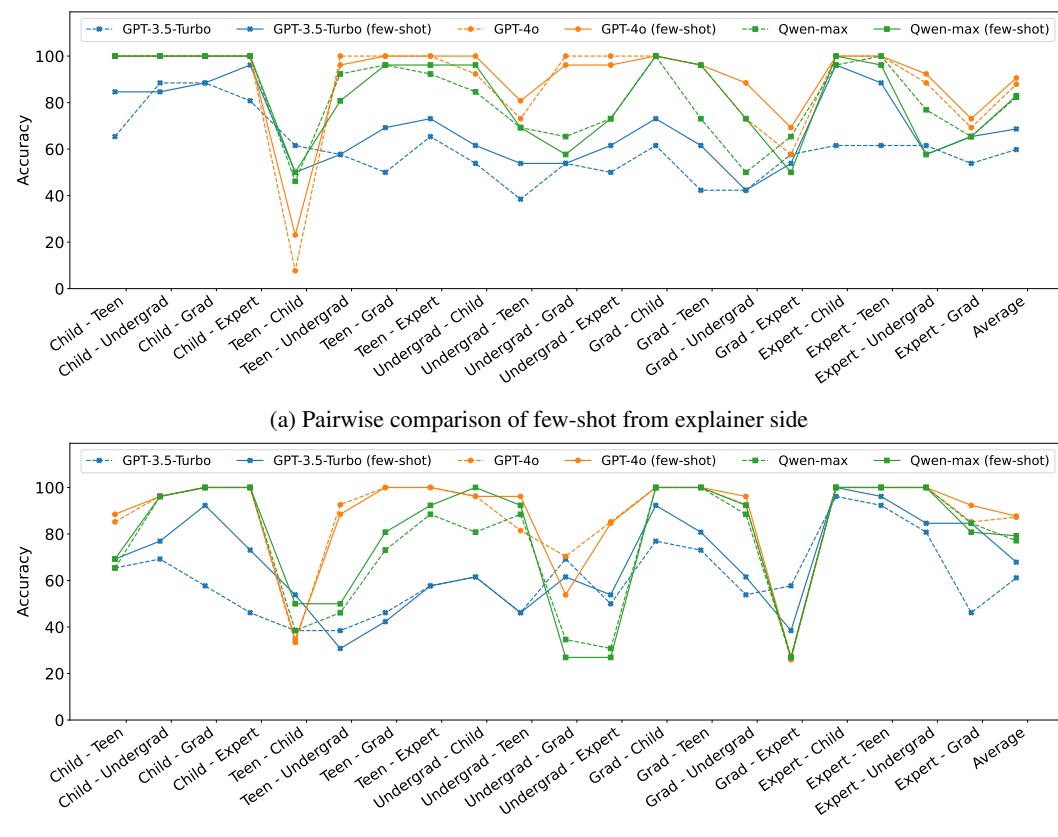

(a) Pairwise comparison of few-shot from explainer side

(b) Pairwise comparison of few-shot from audience side

Figure 5: **Performance of pairwise role comparison with few-shot learning from explainer and audience perspectives.** Few-shot examples generally improve accuracy across all models, with GPT-4o showing the most consistent gains, particularly for pairs with distinct language differences. Qwen-Max benefits to a lesser degree, while GPT-3.5-turbo shows varied results, indicating uneven effects of few-shot learning. Despite these improvements, all models struggle with closely related roles, suggesting that few-shot learning's effectiveness is model-dependent and context-sensitive, especially in nuanced role differentiation tasks.

spectives. In Figure 4, the inclusion of few-shot examples significantly improves both precision and recall for GPT-4o across most role levels, particularly in clearly distinguished roles like "Child" and "Expert". However, for GPT-3.5-turbo and Qwen-Max, few-shot examples have a more inconsistent effect, with some role levels seeing notable improvements while others experience declines or remain unchanged. This variability highlights that while few-shot learning enhances performance for advanced models like GPT-4o, its influence is less predictable for other LLMs, particularly in cases where the role levels are closer or more nuanced.

Figure 5 further explores few-shot learning in the context of pairwise role comparison. Few-shot examples generally enhance accuracy across all models, with GPT-4o showing the most consistent gains, especially for pairs with distinct language differences. Qwen-Max also benefits but to a lesser extent, while GPT-3.5-turbo experiences varied results, indicating an uneven impact of few-shot learning across different role pairs. Despite the overall improvements, all models struggle with specific pairs, such as "Teen - Grad" and "Undergrad - Grad", suggesting that distinguishing between closely related roles remains a challenge even with the additional context provided by few-shot learning. These results suggest that while few-shot learning is beneficial in enhancing LLM evaluators' accuracy, its effectiveness is model-dependent and context-sensitive, particularly for tasks requiring nuanced role differentiation.

## 5 USER STUDY

To evaluate human performance on the tasks defined by our PersonaEval benchmark and to gain insights into the human experience with these tasks, we conducted a comprehensive user study. This study aimed to assess how well humans can classify dialogues according to target levels and to compare their performance with that of large language models like GPT-4o.

### 5.1 SETTINGS

We recruited 15 volunteers, including 10 undergraduates and 5 PhD students from various academic disciplines. The study consisted of two main tasks: (1) Single Answer Role Grading: Participants classified 60 dialogues from our PersonaEval benchmark (30 from the explainer perspective and 30 from the audience perspective) into one of five levels based on complexity and target alignment. (2) Pairwise Role Comparison: The 5 PhD students also compared pairs of dialogues, selecting the one that best matched a target level. Each participant answered 40 questions (20 for each perspective). No time limits were set, allowing participants to complete them at their own pace. This approach was intended to minimize pressure and encourage careful consideration in their responses.

### 5.2 ANALYSIS

In the Single Answer Role Grading Task, participants achieved an average accuracy of 48% on the explainer perspective, closely aligning with GPT-4o's accuracy of 47%. On the audience perspective, participants attained a higher average accuracy of 54%, whereas GPT-4o achieved 67%. The higher accuracy on the audience side suggests that audience dialogues were slightly easier to classify, possibly due to more relatable language or clearer indicators of comprehension. A comparison between educational backgrounds revealed that PhD students outperformed undergraduates in both roles, suggesting that advanced education may improve the ability to recognize subtle differences in dialogue complexity and appropriateness. Importantly, participant accuracy significantly exceeded the chance level of 20%, confirming that humans can identify patterns within the dialogues that enable them to distinguish between levels effectively. This finding underscores the validity of the benchmark and its alignment with human cognitive abilities.

In the Pairwise Role Comparison Task, PhD participants achieved an average accuracy of 87% on the explainer role and 86% on the audience role, closely matching GPT-4o's accuracy of 90% on both roles. The high accuracy rates in this task indicate that when directly comparing dialogues, both humans and the language model can effectively assess alignment with target levels. Despite strong performance, evaluating long dialogue pairs may have posed cognitive challenges for participants, as processing and comparing large amounts of information could lead to occasional inaccuracies. However, human evaluators generally maintained consistency in their judgments across different contexts. In contrast, language models like GPT-4o may exhibit inconsistencies due to the nature of prompt-based evaluations. Since the model assesses the same dialogues with varying prompts for different target levels, it may assign different levels in separate evaluations of the same dialogue pair. This discrepancy arises because the model does not retain context between evaluations, unlike human participants who can integrate information holistically.

## 6 CONCLUSION

We explore the role-playing capabilities of large language models (LLMs) by introducing a benchmark that assesses their ability to evaluate dialogues across varying levels of complexity. Using real-world data from the Wired 5 Levels video series, we introduce PersonaEval with three evaluation settings, single answer role grading, pairwise role comparison, and reference-guided role grading, to comprehensively analyze LLMs' effectiveness in role adherence and comparison tasks. Our results show that while advanced models like GPT-4o demonstrate strong performance, they still face challenges in accurately distinguishing between closely related role levels, highlighting limitations in their evaluation capabilities. Few-shot learning enhances performance, particularly for more advanced models, but its impact varies across LLMs, indicating room for improvement. Our PersonaEval underscores the need for continued research to refine LLM evaluators, with the goal of developing models that can more reliably and accurately assess role-playing consistency.

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

## A   DETAILED EXPERIMENT RESULT OF SINGLE ANSWER ROLE GRADING

Table 5: Detailed five-level classification performance of LLMs on explainer-side context across five turns and all turns.

| Model | Metric | 5 turns | | | | | | all turns | | | | | |
|---|---|---|---|---|---|---|---|---|---|---|---|---|---|
| | | Child | Teen | Undergrad | Grad Student | Expert | Average | Child | Teen | Undergrad | Grad Student | Expert | Average |
| GPT-3.5-Turbo | Precision | 97.1 ± 0.9 | 21.8 ± 2.0 | 19.2 ± 1.6 | 26.5 ± 0.3 | 35.4 ± 2.7 | 40.0 ± 0.4 | 100.0 ± 0.0 | 18.8 ± 2.4 | 25.3 ± 3.1 | 27.6 ± 5.9 | 30.2 ± 4.5 | 40.4 ± 1.6 |
| | Recall | 22.2 ± 0.6 | 24.5 ± 2.3 | 22.1 ± 3.2 | 42.8 ± 1.7 | 49.7 ± 4.7 | 32.3 ± 0.5 | 21.8 ± 3.6 | 12.8 ± 1.8 | 26.9 ± 3.1 | 39.7 ± 4.8 | 47.4 ± 10.1 | 29.7 ± 2.8 |
| GPT-4o | Precision | 91.0 ± 1.3 | 45.4 ± 5.8 | 32.0 ± 3.8 | 30.7 ± 2.7 | 45.2 ± 1.4 | 48.9 ± 2.5 | 98.6 ± 2.0 | 71.4 ± 2.3 | 41.5 ± 3.2 | 32.2 ± 2.3 | 47.3 ± 3.4 | 58.2 ± 1.3 |
| | Recall | 71.9 ± 0.5 | 34.1 ± 6.3 | 33.9 ± 4.9 | 43.4 ± 2.6 | 52.2 ± 1.0 | 47.1 ± 2.4 | 80.8 ± 3.1 | 53.8 ± 8.3 | 33.3 ± 7.3 | 41.0 ± 4.8 | 64.1 ± 1.8 | 54.6 ± 1.7 |
| Qwen-max | Precision | 96.3 ± 1.5 | 25.6 ± 1.4 | 27.4 ± 0.5 | 36.9 ± 2.1 | 56.9 ± 17.5 | 48.6 ± 3.2 | 100.0 ± 0.0 | 25.7 ± 4.6 | 30.8 ± 0.3 | 35.5 ± 4.4 | 40.0 ± 14.1 | 46.4 ± 2.5 |
| | Recall | 31.1 ± 1.3 | 33.3 ± 1.0 | 63.0 ± 1.2 | 54.6 ± 2.5 | 3.8 ± 1.0 | 37.2 ± 0.4 | 24.4 ± 4.8 | 25.6 ± 6.5 | 71.8 ± 1.8 | 46.2 ± 8.3 | 5.1 ± 1.8 | 34.6 ± 0.0 |
| Qwen-plus | Precision | 91.9 ± 1.8 | 30.9 ± 1.3 | 27.2 ± 2.0 | 32.4 ± 2.4 | 48.7 ± 2.7 | 46.2 ± 0.8 | 91.5 ± 1.9 | 29.9 ± 4.2 | 25.3 ± 2.8 | 33.1 ± 4.2 | 42.9 ± 7.4 | 44.5 ± 3.9 |
| | Recall | 57.4 ± 1.5 | 27.7 ± 2.5 | 52.4 ± 5.5 | 45.7 ± 2.5 | 21.0 ± 4.0 | 40.8 ± 1.2 | 43.6 ± 9.6 | 19.2 ± 3.1 | 43.6 ± 3.6 | 51.3 ± 6.5 | 25.6 ± 4.8 | 36.7 ± 4.7 |
| Qwen-turbo | Precision | 33.3 ± 47.1 | 6.5 ± 0.3 | 9.6 ± 0.6 | 25.7 ± 0.6 | 25.6 ± 0.6 | 20.2 ± 9.4 | 0.0 ± 0.0 | 3.7 ± 5.2 | 12.8 ± 4.6 | 25.6 ± 2.0 | 26.5 ± 7.0 | 13.7 ± 1.7 |
| | Recall | 0.2 ± 0.2 | 2.4 ± 0.0 | 16.5 ± 1.4 | 61.1 ± 2.5 | 32.8 ± 2.9 | 22.6 ± 0.2 | 0.0 ± 0.0 | 1.3 ± 1.8 | 14.1 ± 6.5 | 66.7 ± 6.5 | 28.2 ± 6.5 | 22.1 ± 1.6 |
| Deepseek-chat | Precision | 98.5 ± 1.3 | 22.4 ± 1.3 | 15.3 ± 1.1 | 30.0 ± 0.9 | 39.8 ± 1.8 | 41.2 ± 0.4 | 96.3 ± 5.2 | 22.8 ± 2.1 | 16.6 ± 6.4 | 28.1 ± 4.1 | 46.0 ± 7.1 | 42.0 ± 1.6 |
| | Recall | 35.0 ± 2.2 | 20.0 ± 0.7 | 20.7 ± 1.4 | 65.2 ± 3.3 | 28.4 ± 2.4 | 33.9 ± 0.2 | 34.6 ± 3.1 | 12.8 ± 1.8 | 15.4 ± 6.3 | 61.5 ± 8.3 | 44.9 ± 7.3 | 33.8 ± 3.3 |

Table 6: Detailed five-level classification performance of LLMs on audience-side context across five turns and all turns.

| Model | Metric | 5 turns | | | | | | all turns | | | | | |
|---|---|---|---|---|---|---|---|---|---|---|---|---|---|
| | | Child | Teen | Undergrad | Grad Student | Expert | Average | Child | Teen | Undergrad | Grad Student | Expert | Average |
| GPT-3.5-Turbo | Precision | 78.4 ± 1.7 | 41.7 ± 1.0 | 40.5 ± 2.8 | 35.9 ± 1.5 | 57.6 ± 0.5 | 50.8 ± 0.3 | 78.5 ± 2.8 | 46.9 ± 2.3 | 46.4 ± 2.7 | 36.9 ± 5.2 | 55.4 ± 4.0 | 52.8 ± 2.8 |
| | Recall | 52.0 ± 3.1 | 50.8 ± 6.6 | 39.7 ± 4.6 | 26.3 ± 1.2 | 91.5 ± 0.8 | 52.1 ± 0.5 | 37.2 ± 3.6 | 47.4 ± 1.8 | 42.3 ± 3.1 | 32.1 ± 7.3 | 96.2 ± 0.0 | 51.0 ± 3.1 |
| GPT-4o | Precision | 68.4 ± 0.2 | 48.9 ± 3.6 | 59.3 ± 2.2 | 64.8 ± 5.0 | 84.0 ± 2.1 | 65.1 ± 1.5 | 61.0 ± 5.7 | 69.4 ± 3.9 | 69.4 ± 4.9 | 80.3 ± 4.5 | | 69.2 ± 4.0 |
| | Recall | 93.6 ± 0.7 | 36.0 ± 1.3 | 53.0 ± 3.6 | 54.0 ± 2.9 | 76.9 ± 4.1 | 62.7 ± 1.3 | 92.3 ± 0.0 | 44.9 ± 9.6 | 62.8 ± 4.8 | 64.1 ± 6.5 | 83.3 ± 4.8 | 69.5 ± 3.6 |
| Qwen-max | Precision | 77.5 ± 0.9 | 43.9 ± 0.6 | 43.5 ± 1.5 | 38.8 ± 2.1 | 88.5 ± 5.0 | 58.4 ± 1.4 | 67.9 ± 4.1 | 45.8 ± 5.8 | 51.5 ± 2.6 | 52.4 ± 2.7 | 91.7 ± 2.0 | 61.9 ± 1.5 |
| | Recall | 63.6 ± 1.3 | 49.2 ± 0.4 | 72.7 ± 1.2 | 49.8 ± 1.8 | 25.1 ± 4.3 | 52.1 ± 1.3 | 35.9 ± 6.5 | 47.4 ± 7.3 | 82.1 ± 4.8 | 70.5 ± 3.6 | 44.9 ± 10.1 | 56.2 ± 2.3 |
| Qwen-plus | Precision | 74.7 ± 1.8 | 47.7 ± 1.7 | 48.4 ± 2.4 | 64.5 ± 3.1 | 82.8 ± 2.1 | 63.6 ± 0.1 | 69.3 ± 1.9 | 53.2 ± 3.5 | 66.0 ± 5.9 | 75.3 ± 6.1 | 76.9 ± 6.0 | 68.1 ± 4.5 |
| | Recall | 85.9 ± 0.7 | 58.6 ± 1.9 | 47.6 ± 4.3 | 37.8 ± 4.3 | 77.2 ± 2.9 | 61.4 ± 0.5 | 66.7 ± 4.8 | 60.3 ± 10.1 | 67.9 ± 1.8 | 51.3 ± 6.5 | 91.0 ± 1.8 | 67.4 ± 4.1 |
| Qwen-turbo | Precision | 79.3 ± 4.0 | 30.8 ± 0.6 | 34.7 ± 1.2 | 29.1 ± 1.6 | 40.0 ± 1.1 | 42.8 ± 1.6 | 51.1 ± 12.6 | 39.5 ± 0.8 | 37.6 ± 6.7 | 37.3 ± 3.8 | 34.2 ± 1.3 | 40.0 ± 3.9 |
| | Recall | 10.1 ± 3.4 | 59.2 ± 2.6 | 39.4 ± 2.2 | 20.3 ± 0.4 | 66.1 ± 2.2 | 39.0 ± 1.2 | 9.0 ± 3.6 | 51.3 ± 4.8 | 32.1 ± 4.8 | 38.5 ± 5.4 | 56.4 ± 1.8 | 37.4 ± 2.4 |
| Deepseek-chat | Precision | 83.1 ± 1.0 | 41.3 ± 0.8 | 52.7 ± 2.1 | 46.9 ± 2.0 | 82.7 ± 1.8 | 61.3 ± 0.5 | 83.0 ± 2.3 | 50.5 ± 2.2 | 61.8 ± 6.3 | 56.0 ± 3.9 | 70.3 ± 3.3 | 64.3 ± 2.8 |
| | Recall | 52.4 ± 1.1 | 70.9 ± 1.9 | 54.6 ± 0.4 | 58.7 ± 1.2 | 51.5 ± 3.4 | 57.6 ± 0.7 | 38.5 ± 6.3 | 74.4 ± 1.8 | 57.7 ± 5.4 | 62.8 ± 4.8 | 70.5 ± 9.6 | 60.8 ± 3.5 |

## B   PROMPTS

### B.1   SINGLE ANSWER ROLE GRADING

> **5 levels classification of explainer/audience side**
> # Requirement
> You are an expression and content evaluation assistant. Your task is to determine which of the following levels a given passage **is intended for/belongs to**: [Child, Teen, College Student, Grad Student, Expert]. The given passage contains 5/all responses of the speaker from a conversation. Analyze the speaking style and the knowledge level presented in the content to make your judgment. Return the most appropriate level in the following format.
> # Response Format
> ```
> {
> "analysis": "the analysis of the passage",
> "level": "the most appropriate level"
> }
> ```

## B.2 PAIRWISE ROLE COMPARISON

---

**Pairwise role comparison of explainer/audience side**

# Requirement

You are an expression and content evaluation assistant. Your task is to determine which of the following two paragraphs is addressed to/spoken by a Child. The given paragraphs both contain all responses of the speaker from a conversation. Analyze the speaking style and the knowledge level presented in the content to make your judgment. Return the most appropriate paragraph in the following format.

# Paragraph 1

Is that where you fold paper to make different animals, like those?

Nope.

Sure.

Yep.

Yeah.

Wow.

I have seen these before, my friends use these.

Wow.

I think that the people that make them are talented. It's hard. Seeing the stuff that we've made here, I'd bet that they could do rocket ships. Just so much that you can do with them.

# Paragraph 2

Wow.

They weren't cut?

How about like that?

So all the folds are reversible?

Twists?

That was cool.

Origami, I think, is the folding of paper to make anything in general, from 3D things to flat things and I think origami is about turning simple things into complex things and it's all about patterns.

# Response Format

```
{
"analysis": "the analysis of the passage",
"level": "the most appropriate level"
}
```

---

## B.3 REFERENCE-GUIDED ROLE GRADING

---

**Prompt1: Few-shot learning for 5 levels classification explainer side**

# Requirement

You are an expression and content evaluation assistant. Your task is to determine which of the following levels a given passage is intended for: [Child, Teen, College Student, Grad Student, Expert]. The given passage contains 5/all responses of the speaker from a conversation. Analyze the speaking style and the knowledge level presented in the content to make your judgment. Return the most appropriate level in the following format.You can refer to the following examples to help you make the judgment.

# Examples

## Level: Child

What's your conception about what the internet is?

The internet is, physically, these computers that all talk to each other. Billions of computers, in the case of the internet. The internet allows us to do a lot of really, really interesting, what we call applications. You ever think about how that video gets to you over the internet?

Got a favorite movie?

---

Matilda. All right. We're gonna actually build an internet. I've got a couple of things here that I wanna show you, or a couple of toys, actually. Okay, let's pretend that these round balls are computers. And the internet is something that connects them. And right now, the internet is just one communication link. And Matilda is sent over the internet from this computer to your computer. So the internet is a network for carrying information from one computer to another. Now this network here looks pretty simple, doesn't it? Right? It's just one thing. Should we add some more friends in?

## Level: Teen

The underseas cables are so cool! They're these big cables that are laid down by switches. They cross both the Atlantic, the Pacific, the Indian Ocean. So the undersea cables are how the networks in Europe, United States, Asia are all connected together.

That's really what we call the first hop. It's like from your phone, from your tablet, from the computer that you're on, there's no cables coming in. You go over a wireless connection. Wi-Fi is the protocol that allows your computer to talk to that first hop router over a wireless communication link.

Ah, you said they're all in one spot. In fact, they're in lots of spots in Netflix. And so most applications would like to connect you with a server that's close to you. Server is really just a big computer with a lot of memory, a lot of discs that store all the Netflix movies, and also so that you don't have to cross over too many internet links to get from where the server is to the TV or the device in your home.

## Level: College Student

So what do you think that thing is?

That is a really good guess. There's parts of that that are definitely about conducting. This is the inside of a quantum computer. Yeah, this whole infrastructure is all about creating levels that get progressively colder as you go from top to bottom down to the quantum chip, which is how we actually control the state of the qubits.

Yeah like physically colder. So room temperature is 300 Kelvin. As you get down all the way to the bottom of the fridge it's at 10 millikelvin. You ever heard of quantum computing? So that's pretty good. So you mentioned superposition, but you can also use other quantum properties like entanglement. Have you heard of entanglement?

Okay so it's this idea that you have two objects and when you entangle them together they become connected. And then they are sort of permanently connected to each other and they behave in ways that are sort of a system now. So superposition is one quantum property that we use, entanglement is another quantum property, and a third is interference. How much do you know about interference?

## Level: Grad Student

Your research is awesome.

That's near and dear to my heart. Multi-party computations.

Right, and it allows you to prove that you've been behaving honestly, without revealing any of the secrets involved that you use to actually behave honestly. So we of course know that zero-knowledge proofs for NP-complete languages plays such a huge role in cryptography. I'm curious. What was your first experience with Np-completeness like?

When you first start to think about proofs as an interactive game where we're talking to each other, did that make zero-knowledge possible?

## Level: Expert

I'm a huge fan of the work that you did in RCP, the Routing Control Platform being a precursor to software-defined networking and the notion that rather than having protocols actually always compute things, that we could compute things in data centers. I'd be interested if you could sort of just reflect back on that time and sort of the beginnings of SDN and where it's come since then.

Right, you couldn't directly do what you wanted to.

Yeah, do you see the softwarization of the internet as a whole happening?

Right, so some people have called that the flattening of the internet, right? I think it used to be on average, you would go through 10 different networks to get from a source to a destination.

It's totally fascinating to me that we have such an important global infrastructure, and yet, the laws that that govern it tend to be very, very local.

# Response Format
```
{ "analysis": "the analysis of the passage",
"level": "the most appropriate level"
}
```

**Prompt2: Few-shot learning for 5 levels classification audience side**
# Requirement
You are an expression and content evaluation assistant. Your task is to determine which of the following levels a given passage belongs to: [Child, Teen, College Student, Grad Student, Expert]. The given passage contains 5/all responses of the speaker from a conversation. Analyze the speaking style and the knowledge level presented in the content to make your judgment. Return the most appropriate level in the following format.You can refer to the following examples to help you make the judgment.
# Examples
## Level: Child
What's this?
Fancy chandelier.
It's a quantum?
What does it do?
An A.
Zero, one.
Can computers help you with your homework? Your really hard homework?
## Level: Teen
Well, time is kind of strange because it's almost a man-made idea. There is the tangible of, you know, how the Earth revolves around the sun or how we orbit around ourselves, it's almost in a way, does it exist if the way that we measure it is manmade?
It's difficult to talk about something without adding time into it.
Yeah, in physics class.
Do you think that in the near or foreseeable future of humans, as we know ourselves now, will there be a time where we are using these formulas and these concepts in our daily lives?
## Level: College Student
Yeah, and the one thing that really confuses me there, I'm thinking about one of the most basic things we learn, I guess, from Interstellar is that the universe is expanding, or space is expanding. And so I'm thinking how does that square with gravity and electromagnetism, which is like kind of predicated on the density of charges or masses.
And when you're talking about the perspective experience, is that just human subjective experience or actual observation for physics?
When you said that, you know, it's more likely for an egg to smash or for glass to smash, and that's probably because there's so many atoms, so much stuff going on. But I'm thinking if we zoom in on, like a single thing, I guess, do we have variations that are extremely unintuitive because, you know, things can happen in a way that isn't the statistical average?
Back with Einstein, you know, we wondered, does time change with speed? And that's another change with that before, we didn't think possible, but I guess we found out eventually some of the fanciful ideas. I guess it's just tiny sliver of hope that.
## Level: Grad Student
I'm interested in networking, IoT, and sort of what kind of data science you can use with the datasets that you get from such devices. One of the things that I designed before was a IoT pill dispenser, essentially, which pairs with your smartphone, which does facial detection and other computer vision controls and can basically tell who's taking some sensitive medication and make sure that they've taken it correctly.
Well, I think the primary challenge is that for sure, but then an additional challenge is keeping everything configured in the way that you expect it to be configured. So for example, most IoT devices require you, when you're configuring them for you to enter some kind of captive login portal where you connect to a local network that the IoT device produces, and then you

can input your Wi-Fi SSID and password. But then say if you were to change the password or the name of your Wi-Fi network or you move to a new place, then suddenly, everything needs to be reconfigured. 'Cause that's a problem that scales linearly.

Not without some persuasion, but I agree that these data have massive, massive research value. Something I'm interested in with my research is collaborating with people who manage these distributed sensor devices, and then taking advantage of those datasets and comparing them to, say you were interested in doing a research project on how daily rush hour traffic impacts the acoustic landscape of New York City. Figuring out, look, this street next to this school is causing visible ratings above what we mandate. And so there needs to be an intervention here.

## Level: Expert

My main focus right now is quantum error correction and trying to understand this concept of fault tolerance which everybody thinks they know it when they see it but nobody in the quantum case can precisely define it.

I was asking a computer scientist recently where to read about fault tolerance in classical computing. He said oh they don't teach that in computer science classes anymore because the hardware has become so reliable. In a quantum system, when you look at it or make measurements, it can change in a way that's beyond your control. We have the following task, build a nearly perfect computer out of a whole bunch of imperfect parts.

The great power of a quantum computer is also it's Achilles's heel. That it's very very sensitive to perturbations and noise and environmental effects. You're just multiplying your problems if all your doing is adding qubits.

That's the challenge I would say. As much progress as we've made it's a frustration to still be facing it.

# Response Format

```
{
"analysis": "the analysis of the passage",
"level": "the most appropriate level"
}
```

---

**Prompt3: Few-shot learning for pairwise role comparison of explainer/audience side**

# Requirement

You are an expression and content evaluation assistant. Your task is to determine which of the following two paragraphs is addressed to/spoken by a Child. The given paragraphs both contain all responses of the speaker from a conversation. Analyze the speaking style and the knowledge level presented in the content to make your judgment. Return the most appropriate paragraph in the following format. You can refer to the following examples to help you make the judgment.

# Examples

## Level: Child

What's your conception about what the internet is?

The internet is, physically, these computers that all talk to each other. Billions of computers, in the case of the internet. The internet allows us to do a lot of really, really interesting, what we call applications. You ever think about how that video gets to you over the internet?

Got a favorite movie?

Matilda. All right. We're gonna actually build an internet. I've got a couple of things here that I wanna show you, or a couple of toys, actually. Okay, let's pretend that these round balls are computers. And the internet is something that connects them. And right now, the internet is just one communication link. And Matilda is sent over the internet from this computer to your computer. So the internet is a network for carrying information from one computer to another. Now this network here looks pretty simple, doesn't it? Right? It's just one thing. Should we add some more friends in?

# Paragraph 1

Is that where you fold paper to make different animals, like those?

```
Nope.
Sure.
Yep.
Yeah.
Wow.
I have seen these before, my friends use these.
Wow.
I think that the people that make them are talented. It's hard. Seeing the stuff that we've
made here, I'd bet that they could do rocket ships. Just so much that you can do with them.
# Paragraph 2
Wow.
They weren't cut?
How about like that?
So all the folds are reversible?
Twists?
That was cool.
Origami, I think, is the folding of paper to make anything in general, from 3D things to flat
things and I think origami is about turning simple things into complex things and it's all
about patterns.
# Response Format
```
{ "analysis": "the analysis of the passage",
"level": "the most appropriate level"
}
```
```

## C  MORE DISCUSSION ON USER STUDY

Overall, the study supports the effectiveness of the PersonaEval benchmark in evaluating role-playing capabilities. The tasks are challenging yet accessible, providing meaningful performance metrics across different roles and expertise levels. However, the comparable performance between human participants and large language models on the explainer side underscores the challenge of this task; even human evaluators struggle to consistently identify correct role levels based solely on linguistic cues. To gain further insights, we asked participants about the level-specific patterns they had identified and utilized to solve the task. Their responses are summarized below:

- **Explainer Side**: Participants observed that when addressing lower levels like Child and Teen, the explainer employed simpler vocabulary, used concrete examples, and avoided technical language. In contrast, for higher levels such as Grad Student and Expert, the explainer utilized more complex terminology and abstract concepts, providing detailed, in-depth explanations that presumed foundational knowledge of the subject.

- **Audience Side**: For lower levels like Child and Teen, audience responses were typically brief and focused on seeking clarification. As levels progressed to Grad Student and Expert, responses became more detailed, often engaging critically with the content. Higher-level audiences not only demonstrated a deeper understanding but also frequently discussed topics related to their own research, posed complex questions, and provided insights indicative of a sophisticated grasp of the subject matter.

## D  FULL LIST OF THE TOPICS IN PERSONAEVAL

Algorithm; Black Hole (explained by a astronomer); Black Hole (explained by an astrophysicist); Black Hole; Blockchain; Connectome; CRISPR; Dimension; Fractals; Gravity; Hacking; Harmony; Infinity; Internet; Laser; Machine Learning; Memory; Moravec's paradox; Nanotechnology; Nuclear Fusion; Origami; Quantum Computing; Quantum Sensing; Sleep; Time; Virtual Reality; Zero-knowledge proof

