# OpenReview forum: "PersonaEval: Benchmarking LLMs on Role-Playing Evaluation Tasks"
_ICLR.cc/2025/Conference — ICLR 2025 Conference Withdrawn Submission_

### Official Review · Reviewer_Hdru · 2024-10-19

**Soundness:** 2
**Presentation:** 3
**Contribution:** 2
**Rating:** 3
**Confidence:** 4

**Summary:**

- A new benchmark designed to assess the effectiveness of large language models (LLMs) in role-playing evaluation tasks using real-world data from the Wired 5 Levels video series.

- Evaluation Settings: (1) Five-level classification task to determine expertise levels. (2) Pairwise comparison of role-based responses. (3) Reference-guided role grading with few-shot learning.

- Data is drawn from experts explaining complex topics to five distinct audiences (child, teenager, college student, graduate student, expert), testing the models' ability to adapt language to different knowledge levels.

- Current LLMs (e.g., GPT-4, GPT-3.5-turbo) show limitations in accurately evaluating role-playing tasks, particularly in distinguishing closely related roles.

- Incorporating reference examples (few-shot learning) improves model performance but is inconsistent across different models.

**Strengths:**

- While the use of the Wired 5 Levels dataset may not be ideal for all role-playing research, its application in this context is creative and introduces a fresh angle for evaluating how well models adapt to varied linguistic and comprehension levels.

- The evaluation settings—single answer grading, pairwise comparison, and reference-guided grading—are well thought out and align with standard LLM evaluation methodologies. This creates a good framework for assessing model performance in role-play scenarios.

- The paper is clearly structured, and the steps taken in developing PersonaEval are well explained. The use of real-world data and the clear description of different audience levels make it easy to understand the evaluation process and the dataset's relevance, even if it may not be universally ideal for role-playing research.

 - While the Wired 5 Levels dataset might not directly suit all role-playing applications, it provides a robust starting point for research into adaptive language use, making the work potentially useful in related fields like pedagogical AI or communication studies.

**Weaknesses:**

- The link between the Wired 5 Levels dataset and persona role-playing feels underdeveloped. The five levels in the dataset represent different knowledge audiences but not distinct personas. This conflates the complexity of role-playing (adapting personality, tone, or behavior) with a different challenge—adjusting language complexity to suit expertise. Strengthening the theoretical connection between these two aspects is essential to ensure the benchmark addresses role-playing rather than just audience targeting. Clarify why expertise-based adaptation can serve as a proxy for persona role-playing. Alternatively, consider using or supplementing the dataset with interactions where LLMs adopt clearer, more defined personas beyond knowledge levels. Did the authors think of an ablation method?



- The Wired 5 Levels data may be too confounded to yield clear conclusions about role-playing. Since the dataset measures language complexity rather than behavioral adaptation, it becomes challenging to isolate whether LLMs are learning to play a role or simply to adjust based on cognitive ability. This makes it hard to draw strong conclusions about LLM performance in role-playing specifically. Introduce additional experiments that focus explicitly on personas, personality shifts, or role-driven interactions. This could help to better test how well LLMs modulate their responses in line with distinct personas rather than just shifting linguistic complexity. Linguistic complexity is also a rich research field, and there are many linguistic feature extractors and models out there.



- The claim that “most” role-playing research relies on LLM evaluation (lines 037-039) is overstated. While LLMs are widely used for evaluation tasks, the field is more diverse, with various supporting metrics often complementing LLM evaluations.  The role-playing research landscape is more diverse. (Kovač-2024, Lee-2024). Or they use some supporting metrics when they use LLM evaluations (Wang-2024).

(Kovač-2024) "Stick to your role! Stability of personal values expressed in large language models."

(Lee-2024) "Language Models Show Stable Value Orientations Across Diverse Role-Plays."

(Wang-2024) "Inch saracter: Evaluating personality fidelity in role-playing agents through psychological interviews."

**Questions:**

Do you have plans to reframe this research away from role-playing?

Given the nature of the Wired 5 Levels dataset, which focuses on adjusting explanations based on expertise rather than persona, do you intend to reframe the study towards a more fitting evaluation domain, like pedagogical adaptation or communication complexity? This research feels more aligned with tasks that involve adapting information for different audiences rather than role-playing, as it’s commonly understood in the literature.

**Details Of Ethics Concerns:**

Is the Wired video copyrighted?

---

> ### Author Response · Authors · 2024-11-26
>
> Thank you for the thoughtful and detailed feedback. We reply point-by-point here, to begin the discussion.
>
> W1&W2: Our goal is to assess a fundamental aspect of role-playing: the ability of LLMs to adapt language and communication to align with different audiences. While it is true that the Wired 5 Levels dataset primarily involves adjusting language complexity for varying expertise levels, we argue that this is a critical subset of role-playing. Every persona inherently involves some degree of expertise-based adaptation. If LLMs struggle to differentiate between audiences with clear and structured knowledge-level distinctions, it raises deeper concerns about their ability to handle more complex and nuanced persona shifts involving tone, intent, or behavior.
> That said, we agree the current setup may conflate language complexity with behavioral adaptation. To address this:
> 1. We plan to conduct an ablation study to isolate the impact of linguistic complexity, such as controlling for expertise-specific content or randomizing audience indicators, to clarify what LLMs are truly learning.
> 2. Future work will expand the benchmark to include datasets or tasks that explicitly evaluate personality shifts or behavioral role adaptation, providing a broader view of LLM capabilities in role-playing.
> We acknowledge that linguistic complexity itself is a well-studied area and appreciate the suggestion to incorporate advanced linguistic feature analysis. We will explore such tools to further disentangle role-specific behaviors from mere complexity adjustments in future iterations of our work.
>
> W3: We appreciate the reviewer pointing out the overstatement in our claim regarding the prevalence of LLM-based evaluation in role-playing research. While it is true that many works utilize supporting metrics alongside LLM evaluations (e.g., Kovač-2024, Lee-2024, Wang-2024), our intent was to highlight that a significant portion of studies rely primarily on LLM evaluations as the central metric. We acknowledge that our phrasing was imprecise and will revise the manuscript to more accurately reflect this nuance. Thank you for bringing this to our attention.
>
> Q: We sincerely thank the reviewer for this valuable suggestion. Reframing the study toward pedagogical adaptation or communication complexity is indeed a compelling direction, given the nature of the Wired 5 Levels dataset. We will carefully consider this perspective for future iterations of this research. That said, in this work, our primary goal is to assess whether LLMs can distinguish between audiences based on expertise levels, as a foundational test of role-playing ability. Role-playing often involves expertise-based linguistic adaptation as one of its core components. If LLMs struggle with even this straightforward task, it suggests broader challenges in simulating more nuanced personas that require behavioral shifts, tone adjustments, or emotional intelligence. We will revise the manuscript to acknowledge the alternative framing and clarify how expertise-based adaptation serves as a stepping stone for evaluating the broader capabilities of LLMs in role-playing.

---

### Official Review · Reviewer_g62X · 2024-10-20

**Soundness:** 2
**Presentation:** 3
**Contribution:** 2
**Rating:** 3
**Confidence:** 3

**Summary:**

This paper introduces a benchmark designed to assess the capabilities of language models in evaluating role-playing performances, specifically focusing on their ability to discern and classify different levels of communication expertise. So, **it is about the meta-evaluation**. The benchmark utilizes transcripts from the Wired 5 Levels video series, where experts explain complex concepts to audiences of varying educational levels (child, teen, college student, graduate student, and expert).

The central premise is that a model's ability to classify these educational levels in dialogue could be **a proxy** for its potential effectiveness in evaluating role-playing models.

The paper presents three evaluation settings: single-answer role classification, pairwise role comparison, and reference-guided role classification. These settings test the model’s capability to distinguish and evaluate role-specific language.

One of the paper's goals is to provide insights that could improve the professionalism and precision of LLM-based evaluators in role-playing contexts.

**Strengths:**

1. The topic of role-play meta-evaluation is indeed a gap in current research.
2. Applying the Wired 5-level dataset for a task used as a proxy for role-play meta-evaluation is original.
3. The paper is well-structured and generally clear in its presentation.
4. It is good that three different evaluation settings (single-answer role grading, pairwise role comparison, and reference-guided role grading) are included. This comprehensive approach aligns well with various evaluation methodologies used in existing role-playing benchmarks.
5. Incorporating human performance comparisons adds value to the study. By providing this baseline, the paper offers crucial context for interpreting the models' results.

**Weaknesses:**

1. The paper does not support the central premise (see the summary section). It is implicitly assumed true throughout and might benefit from more explicit justification. The paper does not adequately address how well the performance on this benchmark might transfer to evaluating role-playing models. I'm unsure how to fix it, as it seems to be a global framing problem. The authors should probably add more datasets from different sources. Some potential problems are:
    * 1.1. The domain of conversations is specific to educational videos, which may limit generalizability to broader role-playing scenarios. It may introduce biases or limitations in the types of language and expertise levels represented.
    * 1.2. Role alignment is only one criterion out of many possible criteria for role-play evaluation.
    * 1.3. In the actual LLM role-playing conversation, one side is LLM. In the proposed task, both sides are humans.
    * 1.4. Archetypes are not roles. There are no detailed descriptions for them.
2. The paper could benefit from acknowledging and comparing with [the Judgemark benchmark](https://eqbench.com/judgemark.html), which assesses evaluators for creative writing without proxy tasks.
3. The supplementary code is poorly organized, and the results are challenging to reproduce. There are no instructions on how to run the scripts to get the results, and some comments and outputs are in Chinese. I’ve tried to reproduce some numbers from Table 2 (GPT-4o, 5 turns), and they are considerably different from the ones stated in the paper, outside of the confidence intervals stated in Appendix A.
4. Dataset collection methodology is not stated. As far as I can see, there is no explanation of how the videos were converted into a clean text dataset. I found at least two other papers using the same data source: ["A Dialogue Corpus for Learning to Construct Explanations"](https://aclanthology.org/2022.coling-1.27/) and ["Evaluating the Explanation Capabilities of LLMs in Conversation Compared to a Human Baseline"](https://arxiv.org/abs/2406.18512). None of them are cited in this paper.
5. Tables 2, 3, and 4 contain too much information. The numbers should support some hypothesis. Most of the numbers in these tables are unclear about why they are there or what hypothesis they support. Consider replacing precision/recall with f-measure and reporting only one number per model. Full tables can still be included in the appendices.

I'm leaning toward rejecting the paper. Every weakness is fixable except #1, which prevents the paper from being useful for its own goals.

**Questions:**

### Questions
1. How exactly the dataset was obtained?
2. Why do you think your benchmark is general enough to be used for role-play meta-evaluation?
3. How are the users of your benchmark?
4. How are you going to support your benchmark? Are you going to support it at all?

### Suggestions
1. Please include an analysis of the costs for running these evaluations. Those are tracked in W&B and should be relatively easy to add.
2. The name "PersonaEval" might be reconsidered to reflect better the benchmark's focus on evaluating evaluators rather than personas directly. Also, "CharacterEval" is already out there, and it is easy to be confused with.
3. Line 530: The first sentence in the conclusion could be rephrased to more accurately reflect the paper's focus on evaluating LLMs' ability to assess dialogues rather than exploring role-playing capabilities directly.
4. Table 1: You should round the average number of tokens to the nearest integer. For instance, 585.23 -> 585. ".23" doesn't add any useful information.
5. Only three families of models are covered. You should probably add at least one more.
6. Fix the tables (weakness #5).
7. Appendix C should be in the main body of the paper. It explains linguistic clues that humans and models use to get their predictions.

**Details Of Ethics Concerns:**

Two things:
1. I'm concerned about the dataset used in this paper. In [this paper](https://arxiv.org/abs/2404.10475), authors explicitly obtain permission from WIRED to publish findings on their data, which is the right thing to do. Also, it should be explicitly stated if some pre-processing procedures from other papers were used (or not used).
2. This study involved 15 volunteers. What was the procedure for recruiting them? I guess they were not paid, so what was their motivation? This can be perfectly fine, but authors should be explicit about it.

---

> ### Author Response · Authors · 2024-11-26
>
> Thank you for the thoughtful and detailed feedback. We reply point-by-point here, to begin the discussion.
>
> W1: We believe our benchmark serves as a necessary condition for evaluating LLMs as evaluators in role-playing contexts, focusing specifically on their ability to distinguish different levels of expertise. While the educational video domain may not fully capture every aspect of role-playing, we argue that it provides a structured and controlled environment for testing the core evaluative capabilities of LLMs, which can be extended to other domains with further validation. Additionally, we will discuss potential future work, including the incorporation of more diverse datasets to better capture the complexities of role-playing evaluations.
>
> W2: Thank you for the suggestion. While creative writing evaluation differs from our focus on role-playing, we will add a discussion of the Judgemark benchmark to the related work section.
>
> W3: We will reorganize the code to improve clarity and ensure it is easier to reproduce.
>
> W4 & Q1: The Wired 5 Levels dataset was sourced from the Wired 5 Levels video series, where experts explain a concept to five distinct audiences. We manually segment and refine the transcriptions for audiences at different levels, and separate the turns of explainers and audiences. We will cite the relevant papers you mentioned to acknowledge their contributions.
>
> W5: We will reorganize the tables to focus on key metrics that directly support our hypotheses, such as reporting the f-measure for each model.
>
> Q2: While our benchmark specifically evaluates expertise-based adaptation, we argue that this is a core component of role-playing, which often involves adjusting language, tone, and complexity based on the audience. By testing LLMs' ability to distinguish between different levels of expertise, we assess how well LLMs can modulate responses in line with role-specific communication needs. Although it focuses on a subset of role-playing, our benchmark lays the groundwork for evaluating more complex role-play scenarios in future work, offering insights into LLM performance across different domains.
>
> Q3: The primary users of our benchmark are researchers and practitioners interested in evaluating and improving LLMs in the context of role-playing and audience-specific communication. This includes those working on human-computer interaction, natural language understanding, and applications where LLMs must adapt to diverse user personas. Additionally, it is aimed at those developing evaluation metrics for LLMs, offering a structured way to assess language adaptation across varying audience types.
>
> Q4: We plan to support and maintain the benchmark by releasing it publicly and providing clear documentation, including detailed instructions on how to use it for various evaluation tasks. Additionally, we will continue to improve the benchmark by incorporating user feedback, updating it with new datasets and tasks that test broader aspects of role-playing (e.g., emotional tone, intent), and potentially integrating additional evaluation methods. We aim to foster a community of researchers who can contribute to and expand upon the benchmark, ensuring it evolves alongside advancements in LLM evaluation.
>
> Suggestions: We appreciate the reviewer’s constructive suggestions. We plan to incorporate points 1, 4, 5, and 6 into the revised manuscript. Regarding point 7, while we recognize the value of this suggestion, we are somewhat constrained by the limited space in the paper and will address it to the extent possible within these constraints. For point 3, we will revise our wording for greater clarity, while reiterating that the primary focus of our paper remains on evaluating LLMs’ ability to assess dialogues based on audience expertise. We are grateful for the reviewer’s feedback and will make the necessary adjustments accordingly.

---

> > ### Comment · Reviewer_g62X · 2024-11-26
> >
> > Thank you for the response.
> >
> > > W1: We believe our benchmark serves as a necessary condition for evaluating LLMs as evaluators in role-playing contexts, focusing specifically on their ability to distinguish different levels of expertise. While the educational video domain may not fully capture every aspect of role-playing, we argue that it provides a structured and controlled environment for testing the core evaluative capabilities of LLMs, which can be extended to other domains with further validation.
> >
> > Why do you believe in that? What specific aspects of role-playing are captured by the educational video domain? The connection between classifying transcripts of educational videos and role-playing appears tenuous, as it seems to rely only on intuitive rather than demonstrated links.
> >
> > > Q2: While our benchmark specifically evaluates expertise-based adaptation, we argue that this is a core component of role-playing, which often involves adjusting language, tone, and complexity based on the audience. By testing LLMs' ability to distinguish between different levels of expertise, we assess how well LLMs can modulate responses in line with role-specific communication needs. Although it focuses on a subset of role-playing, our benchmark lays the groundwork for evaluating more complex role-play scenarios in future work, offering insights into LLM performance across different domains.
> >
> > Again, it is very vague. Can you prove any of these points?
> >
> > > W4 & Q1: The Wired 5 Levels dataset was sourced from the Wired 5 Levels video series, where experts explain a concept to five distinct audiences. We manually segment and refine the transcriptions for audiences at different levels and separate the turns of explainers and audiences. We will cite the relevant papers you mentioned to acknowledge their contributions.
> >
> > If you were not using datasets processed in those papers, then it's fine not to cite them. But then, how did you get the original transcriptions? What software did you use to transcribe? Can you describe the whole process, from the source videos to the final refined transcriptions with separate turns?

---

### Official Review · Reviewer_QT6q · 2024-11-02

**Soundness:** 3
**Presentation:** 3
**Contribution:** 2
**Rating:** 5
**Confidence:** 4

**Summary:**

The paper introduces PersonaEval, a benchmark designed to assess LLMs' effectiveness in evaluating role-playing through a classification task, utilizing data from the Wired 5 Levels series, where experts explain concepts to different audience types (child, teenager, college student, graduate student, and expert). The paper evaluates GPT-4, GPT-3.5-turbo, and various models from the Qwen family. The paper highlights limitations in current LLMs' ability to evaluate nuanced role-playing tasks.

**Strengths:**

+ Important research question, since many papers are using LLMs as evaluators.
+ Dataset constructed using real-world conversations from a famous TV series, including diverse topics and conversations from people of 5 different levels of knowledge.
+ Two aspects of using the dataset, i.e., the classification, and the comparison, with each aspect having two settings of w/ or w/o demonstrations.

**Weaknesses:**

- The findings may not generalize to every LLM evaluators since the dataset is limited in its scope (first paragraph in the questions).
- There could be some deeper analysis like superficial correlation (shortcut learning). (question 8 & 9)
- Need further elaboration on settings (question 4 5 6 7)
- Need explanation about findings in this paper and related work (question 10)

**Questions:**

Given that many papers are using LLMs for evaluation, it is important to have a comprehensive evaluation on how LLMs can serve as evaluators. However, as I read the paper, I realize that the scope is limited to personas with different knowledge levels. I think the findings in this paper may not generalize to personas having nuanced personality traits, such as different levels of extraversion. The authors could add a discussion about this limitation in the paper.

I have some questions:

1.	Table 1: Is the “Avg Tokens” averaged on the scale of the whole dialogue or just one turn? It looks a bit weird that Child has a higher Avg Turns but lower Avg Tokens.
2.	For Table 2 & 3, could you please provide overall accuracy for each model?
3.	Could you provide the full list of the 26 topics in the appendix? I do take a look on the TV series website, but I cannot see clearly what the topics are.
4.	How many times do you run one LLM to obtain mean and std? What temperature, top_p, and other parameters do you set for LLMs?
5.	For pairwise evaluation, what is the difference between using Child – Teen and Teen – Child? Do they use the same dialogues but different questions (e.g., which is from the child/teen)?
6.	For pairwise evaluation, do you use dialogues in a same topic? If we select a child’s speaking from topic A while a teen’s speaking from topic B, how will LLMs perform?
7.	For pairwise evaluation, since the 5 levels (child, teen, undergrad, graduate, expert) are ordinal data, we can just ask LLMs which response shows the higher level of knowledge. This setting can do some interesting, such as we select two teen’s responses and ask LLMs which one is more knowledgeable (probably providing an option to indicate a tie).
8.	For pairwise evaluation, especially in the contradiction analysis (Table 4): since the contradiction rates for LLMs are relatively high, I am thinking is it because that there are some positional biases? E.g., the order of the two responses.
9.	Related to the previous one: I think it will be interesting if we can have some analysis on some superficial features, such as whether the length of the dialogue can indicate the speaker.
10.	Since the paper claims a limited performance of LLMs being evaluators, there is an inconsistency with findings in existing papers. For example, in the Wang et al., 2024b in your reference, they did a human study on checking whether GPT-4 can provide human-like judgment and found the correlation is high. Could you please explain the reason why you can show lower performance in LLMs? In other words, what are the flaws in existing papers as you mention: “Current approaches to using LLMs in role-playing evaluations are not without flaws.”
11.	I wonder how LLMs can simulate conversations of different expertise levels, that is, we instruct LLMs to speak and act like a teen, or a graduate student. Could you please discuss how your dataset can help in evaluating this aspect?

Overall, the presentation is good in this paper. There are still some minor issues:

1.	Line 88 “large language models (LLMs)” -> “LLMs” since the abbreviation has appeared before. Same as Line 257: “large language models” -> “LLMs”.
2.	Figure 2 is not referenced in the main text.

---

> ### Author Response · Authors · 2024-11-26
>
> Thank you for the thoughtful and detailed feedback. We reply point-by-point here, to begin the discussion.
>
> W1: The ability to accurately classify the speaker’s level is a necessary condition for ensuring that the model can evaluate whether the role-playing output is appropriate, consistent, and well-suited to the target character. This capability is essential for assessing whether the role-playing aligns with the intended role’s complexity, tone, and depth.
>
> Q1: The "Avg Tokens" are calculated based on the entire dialogue. The lower average tokens for the "Child" role arise because both the child's replies and the explainer's responses are shorter, as explanations tailored for children are generally more concise and simplified.
>
> Q2: The "recall" values presented in Tables 2 and 3 can be interpreted as accuracy, as they reflect the overall performance of each model in correctly classifying the roles.
>
> Q3: We have added the full list of the 26 topics in the appendix.
>
> Q4: Due to limited computational resources, we conducted each experiment three times to calculate the mean and standard deviation. For the LLM settings, we used a temperature of 0.7 and top_p of 0.8, with all other parameters set to their default values. We will clarify this in the revised manuscript to ensure reproducibility.
>
> Q5: The pairwise evaluations use the same dialogues but with different questions, such as distinguishing which response is from the child or the teen.
>
> Q6: The pairwise evaluation uses dialogues from the same topic to ensure a fair comparison. Evaluating LLM performance when comparing responses across different topics is an interesting direction and will be explored in future work.
>
> Q7 & Q9: The cases with responses from the same level (e.g., two teen responses with a tie option), would be an interesting extension. We will explore this setting and include an analysis of superficial features as part of our future work.
>
> Q8: The order of the two responses is randomized for each evaluation to minimize positional bias. Despite this randomization, the LLMs still exhibit high contradiction rates, indicating that the issue is not solely due to order effects. That said, we acknowledge the value of further analysis and plan to conduct an ablation study to explicitly investigate the influence of order on the results.
>
> Q10: Our benchmark evaluates LLMs as evaluators in role-playing tasks from a different perspective compared to prior work. Our benchmark specifically assesses the ability of LLMs to classify nuanced roles based on linguistic cues, such as expertise levels. This task introduces unique challenges that may not be addressed in previous evaluations, highlighting areas where current approaches may fall short.
>
> Q11: While simulating conversations at different expertise levels is an interesting aspect, it is not the primary focus of our paper. Our work is centered on evaluating how LLMs assess role-playing performances across different expertise levels, rather than role-playing on different levels.
>
> Typos: We have fixed these errors in the revised manuscript.

---

> > ### Comment · Reviewer_QT6q · 2024-11-28
> >
> > Thanks for your detailed response. It really adds great clarity to your paper. However, my main concern, the W1 (limited scope), is not fully addressed. Maybe the paper can benefit from some restricting the topic to Knowledge Level Evaluation. Currently, PersonaEval may indicate readers that the work can evaluate LLMs' ability to distinguish between different personas, which is not included in the paper.
> >
> > Another remaining question is Q9. Sorry for not expressing clearly. I think Q9 is different from Q7 and should not be answered jointly. I was wondering whether there are length biases, i.e., models tend to choose longer texts for more knowledgable persona.

---

### Official Review · Reviewer_MdsD · 2024-11-04

**Soundness:** 2
**Presentation:** 2
**Contribution:** 2
**Rating:** 5
**Confidence:** 3

**Summary:**

- This work introduces a benchmark designed to evaluate whether large language models (LLMs) can distinguish sentences by speaker expertise levels using only linguistic cues, utilizing data from the Wired 5 Levels video series.
- It presents three evaluation settings specifically tailored to assess LLM performance in role-playing tasks.
- The study finds that GPT-4o achieves the highest performance overall; however, distinguishing roles based on explanation-focused cues is more challenging than from the listener's perspective. Additionally, the amount of contextual information provided significantly impacts the accuracy of role differentiation.

**Strengths:**

- The authors effectively motivated the issues surrounding the reliability of LLM evaluators in role-playing tasks.
- This work covered three comprehensive evaluation settings—single-answer grading, pairwise comparison, and reference-guided evaluation—aligning well with established LLM evaluation methodologies.

**Weaknesses:**

- The Wired 5 Levels video series emphasizes informative content, leading the benchmark to focus primarily on informative conversations. The coverage may be limited.
- While the authors argue that familiar character archetypes reduce performance risks from role unfamiliarity, this approach may also limit the benchmark’s validity in more diverse user scenarios/user types.
- The related work section on role-playing evaluation could more directly address specific issues and contributions within role-playing tasks, as it currently focuses on general LLM evaluation methods.
- This work showed that distinguishing roles using explanation-focused cues is challenging. This can be because LLMs may rely more on language style than on informational depth. A detailed analysis of benchmark sentence differences between explanation and listener-focused parts could clarify these findings.
- The relatively low performance observed in user study raises questions about the task's suitability. While distinguishing between roles like "graduate student" and "undergraduate student" might imply an ability to detect subtle differences, it could also reflect inherent biases about the expected characteristics of each group. This leads to a critical question: is high performance in this task genuinely necessary to serve as an effective evaluator in role-playing tasks?

**Questions:**

- How does accurately predicting the speaker's level demonstrate reliability and effectiveness when using LLMs as role-playing evaluators?
- Given the 26 topics, were there any notable differences in performance by topic?

---

> ### Author Response · Authors · 2024-11-26
>
> Thank you for the thoughtful and detailed feedback. We reply point-by-point here, to begin the discussion.
>
> W1: Our benchmark is designed to assess the effectiveness of LLMs in role-playing evaluation tasks by framing the problem as a classification task. Informative content is well-suited for this purpose, as it provides clear linguistic cues necessary for distinguishing between sentences from different levels of expertise. In contrast, casual conversations lack the structured features required for accurate role differentiation, making them misaligned with our current objective.
>
> W2: Using the five familiar educational levels ensures that the task remains well-defined and feasible for LLMs, as these roles provide clear and structured linguistic distinctions. Expanding to more diverse or unfamiliar user types could introduce ambiguities, making the classification task less reliable and harder to evaluate.
>
> W3:  While we aimed to provide a broader context by discussing general LLM evaluation methods, we agree that focusing more on role-playing-specific challenges and prior contributions would strengthen the section. We will revise this part to better highlight the unique aspects of role-playing evaluation and how our work builds upon or differs from prior studies.
>
> W4: Our benchmark focuses on classifying roles based on linguistic cues without restricting the analysis to explanation-focused cues. Both language style and informational depth are available for LLMs to consider during the classification process. We agree that a deeper analysis of how these factors contribute to the model's decisions could provide additional insights and will consider including this in future work.
>
> W5: The user study results highlight the inherent variability in human judgment, which aligns with our observations regarding the challenges in evaluating nuanced differences. The classification task we designed serves as a simplified yet meaningful scenario to evaluate LLMs' ability to assess role-playing performance. By isolating linguistic cues, this task provides a controlled environment to test whether an LLM evaluator can detect and differentiate persona-specific language effectively. Our benchmark served as a necessary condition for role-playing evaluations, as these tasks require assessing coherence and consistency with a persona in specific contexts. Therefore, we argue that high performance in such classification tasks is essential to developing robust LLM evaluators capable of reliably judging role-playing tasks.
>
> Q1:  Accurately predicting the speaker’s level is crucial for evaluating the effectiveness of LLMs as role-playing evaluators because it reflects their ability to assess the coherence and alignment of role-playing outputs with the intended roles. This demonstrates whether the model can distinguish nuanced linguistic and contextual differences that define role consistency, which is central to evaluating role-playing performance.
>
> Q2: There were differences in performance across topics. To avoid bias from the single topic, we report the average performance across all topics.

---

### Note · Authors · 2025-08-18

I have read and agree with the venue's withdrawal policy on behalf of myself and my co-authors.

---

### Meta-Review · Area_Chair_CuQ8 · 2024-12-19

**Metareview:**

The paper uses Wired 5 levels video series - where an expert attempts to explain a concept to people at 5 different levels of expertises - as an LLM eval. While the premise is interesting, all reviewers agree on a rejection. There are fundamental flaws in the paper's claims (especially regarding their ability to generalize to all role playing agents) that can are made from such a limited dataset

**Additional Comments On Reviewer Discussion:**

While the authors attempted to answer the reviewers' concerns - it not appear to have been done satisfactorily enough to reverse a unanimous rejection.

---

### Decision · Program_Chairs · 2025-01-22

Reject